# Beta R-CNN: Looking into Pedestrian Detection from Another Perspective

**Zixuan Xu**[*]
Peking University
zixuanxu@pku.edu.cn

**Banghuai Li** [*]
Megvii Research
libanghuai@megvii.com

**Ye Yuan**
Megvii Research
yuanye@megvii.com

**Anhong Dang**
Peking University
ahdang@pku.edu.cn

## Abstract

Recently significant progress has been made in pedestrian detection, but it remains challenging to achieve high performance in occluded and crowded scenes. It could be attributed mostly to the widely used representation of pedestrians, *i.e.*, 2D axis-aligned bounding box, which just describes the approximate location and size of the object. Bounding box models the object as a uniform distribution within the boundary, making pedestrians indistinguishable in occluded and crowded scenes due to much noise. To eliminate the problem, we propose a novel representation based on 2D beta distribution, named Beta Representation. It pictures a pedestrian by explicitly constructing the relationship between full-body and visible boxes, and emphasizes the center of visual mass by assigning different probability values to pixels. As a result, Beta Representation is much better for distinguishing highly-overlapped instances in crowded scenes with a new NMS strategy named BetaNMS. What's more, to fully exploit Beta Representation, a novel pipeline Beta R-CNN equipped with BetaHead and BetaMask is proposed, leading to high detection performance in occluded and crowded scenes. Code will be released at *github.com/Guardian44x/Beta-R-CNN*.

## 1 Introduction

Pedestrian detection is a critical research topic in computer vision field with various real-world applications such as autonomous vehicles, intelligent video surveillance, robotics, and so on. During the last decade, with the rise of deep convolutional neural networks (CNNs), great progress has been achieved in pedestrian detection. However, it remains challenging to accurately distinguish pedestrians in occluded and crowded scenes.

Although extensive methods have been attempted for occlusion and crowd issues, the performance is still limited by pedestrian representation, *i.e.*, 2D bounding box representation. The axis-aligned minimum bounding box is widely utilized to explicitly define a distinct object, with its approximate location and size. Although box representation has advantages such as parameterization- and annotation-friendly as the identity of an object, some nonnegligible drawbacks are limiting the performance of pedestrian detection especially in occluded and crowded scenes. Firstly, the bounding box can be regarded as modeling the object as a uniform distribution in the box, but it actually goes against our intuitive perception. Given an occluded pedestrian, what attracts our attention should be the visible part rather than the occluded noise. Secondly, based on box representation, intersection

---

[*]These authors contributed equally

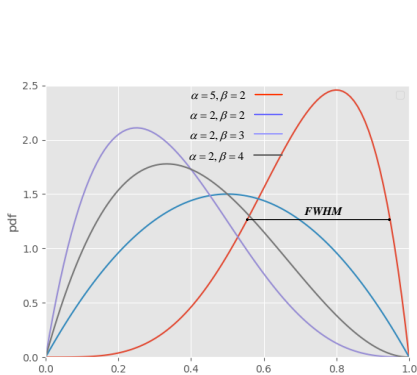

Figure 1: Beta distributions have flexible shapes with different peaks and FWHMs.

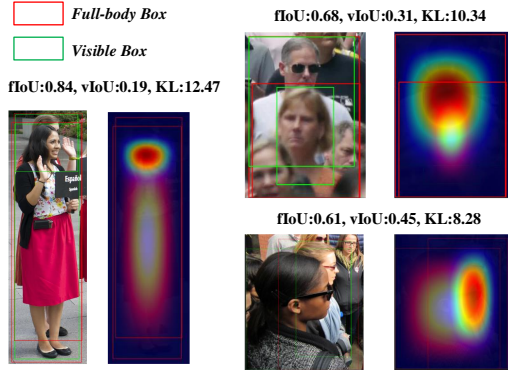

Figure 2: Beta Representation samples and comparisons between IoU and KL divergence.

over union (IoU) serves as the metric to measure the difference between objects, which results in difficulty to distinguish highly-overlapped instances in crowded scenes. As shown in Fig. 2, even if the detectors succeed to identify different human instances in a crowded scene, the highly-overlapped detections may also be suppressed by the post-processing of non-maximum suppression (NMS). Last, the full-body and visible boxes treat a distinct person as two separate parts, which omit their inner relationship as a whole and lead to difficulty for model optimization.

To eliminate the weaknesses of box representation and preserve its advantages in the meanwhile, we propose a novel representation for pedestrians based on 2D beta distribution, named **Beta Representation**. In probability theory, the beta distribution is a family of continuous probability distribution defined in the interval [0, 1], as depicted in Fig. 1. By assigning different values to $\alpha, \beta$, we could control the shape of the beta distribution, especially the peak and the full width at half maximum (FWHM), which is naturally suitable for pedestrian representation with unpredictable visible patterns. We take each pedestrian as a 2D beta distribution on the image and generate eight new parameters as the Beta Representation. As illustrated in Fig. 2, the boundary of 2D beta distribution is consistent with the full-body box, while the peak along with FWHM depends on the relation between the visible part and full-body box. Compared with paired boxes, *i.e.*, full-body and visible boxes, 2D beta distribution treats each pedestrian more like an integrated whole and emphasizes the object center of visual mass meanwhile.

Besides, instead of IoU, Kullback-Leibler (KL) divergence is adopted as a new metric to measure the distance of two objects and the beta-distribution-based NMS strategy is named BetaNMS. Fig. 2 illustrates that while the bounding boxes are too close to distinguish (fIoU > 0.5, vIoU > 0.3[2]), the 2D beta distributions still maintain high discrimination (KL > 7) between each other, thereby leading to better performance in distinguishing highly-overlapped instances.

Moreover, to fully exploit Beta Representation in pedestrian detection, we design a novel pedestrian detector named Beta R-CNN, equipped with two different key modules, *i.e.*, BetaHead and BetaMask. BetaHead is utilized to regress the eight beta parameters and the class score, while BetaMask serves as an attention mechanism to modulate the extracted feature with beta-distribution-based masks. Experiments on the extremely crowded benchmark CrowdHuman [1] and CityPersons [2] show that our proposed approach can outperform the state-of-the-art results, which strongly validate the superiority of our method.

## 2  Related Work

**Pedestrian Detection.** Pedestrian detection can be viewed as object detection for the specific category. With the development of deep learning, CNN-based detectors can be roughly divided into two categories: the two-stage approaches [3, 4] comprise separate proposal generation followed by classification and regression module to refine the proposals; and the one-stage approaches [5–7] perform localization and classification simultaneously on the feature maps without the separate

proposal generation module. Most existing pedestrian detection methods employ either the single-stage or two-stage strategy as their model architectures.

**Occlusion Handling.** In pedestrian detection, occlusion leads to misclassifying pedestrians. A common strategy is the part-based approaches [8–11], which ensemble a series of body-part detectors to localize partially occluded pedestrians. Also some methods train different models for most frequent occlusion patterns [12, 13] or model different occlusion patterns in a joint framework [14, 15], but they are all just designed for some specific occlusion patterns and not able to generalize well in other occluded scenes. Besides, attention mechanism has been applied to handle different occlusion patterns [9, 16]. MGAN [16] introduces a novel mask guided attention network, which emphasizes visible pedestrian regions while suppressing the occluded parts by modulating extracted features. Moreover, a few recent works [17, 18] have exploited to utilize annotations of the visible box as extra supervisions to improve pedestrian detection performance.

**Crowdness Handling.** As for crowded scenes, except for the misclassifying issues, crowdedness makes it difficult to distinguish highly-overlapped pedestrians. A few previous works propose new loss functions to address the problem of crowded detections. For example, OR-CNN [8] proposes aggregation loss to enforce proposals to be close to the corresponding objects and minimize the internal region distances of proposals associated with the same objects. RepLoss [19] proposes Repulsion Loss, which introduces extra penalty to proposals intertwined with multiple ground truths. Moreover, some advanced NMS strategies [20–23, 18] are proposed to alleviate the crowded issues to some extent, but they still take IoU as the metric to measure the difference between detected objects, which limits the performance on identifying highly-overlapped instances from crowded boxes.

**Object Representation.** In computer vision, object representation is one primary topic, and there are many representations for objects in 2D images, such as 2D bounding boxes [4], polygons [24], splines [25], and pixels [26]. Each has strengths and weaknesses from a specific application's practical perspective, providing annotation cost, information density, and variable levels of fidelity. Distribution-based representation has also been tried in [27] which utilizes the bivariate normal distribution as the representation of objects. However, when transformed from bounding boxes rather than segmentation, the mean and variance of bivarite normal distribution are still consistent with the center and scale. Besides, its performance is considerably poor compared to other methods.

In this paper, Beta Representation provides a more detailed representation for occluded pedestrians, along with a new metric to substitute for IoU and a new detector Beta R-CNN, thereby alleviating the occlusion and crowd issues to a great extent.

## 3 Method

In this section, we first introduce the parameterized Beta Representation for pedestrians. Then to fully exploit the Beta Representation, a novel pipeline Beta R-CNN is proposed. Moreover, a specific NMS strategy based on beta distribution and KL divergence, *i.e.*, BetaNMS, is analyzed in detail.

### 3.1 Beta Representation

#### 3.1.1 Beta Distribution

In probability theory and mathematical statistics, the beta distribution is a family of one-dimensional continuous probability distribution defined in the interval $[0, 1]$, parameterized by two positive shape parameters $\alpha$ and $\beta$. For $0 \leq x \leq 1$ and shape parameters $\alpha, \beta > 0$, the probability density function (PDF) of beta distribution is a exponential function of the variable $x$ and its reflection $(1 - x)$ as follows:

$$
\begin{aligned}
Be(x; \alpha, \beta) &= \frac{\Gamma(\alpha + \beta)}{\Gamma(\alpha)\Gamma(\beta)} \cdot x^{(\alpha-1)}(1 - x)^{(\beta-1)} \\
&= \frac{1}{B(\alpha, \beta)} \cdot x^{(\alpha-1)}(1 - x)^{(\beta-1)},
\end{aligned}
\tag{1}
$$

where $\Gamma(z)$ is the gamma function and $B(\alpha, \beta)$ is a normalization factor to ensure the total probability is 1. Some beta distribution samples are shown in Fig. 1. According to the above definition, the mean

$\mu$, variance $\sigma^2$ and shape parameter $\nu$ can be formulated as follows:

$$\mu = E(x) = \frac{\alpha}{\alpha + \beta},$$

$$\sigma^2 = E(x - \mu)^2 = \frac{\alpha\beta}{(\alpha + \beta)^2(\alpha + \beta + 1)}, \qquad (2)$$

$$\nu = \alpha + \beta.$$

### 3.1.2 Beta Representation for Pedestrian

As introduced in Sec. 3.1.1, the beta distribution has two key characteristics: 1) Boundedness, the beta distribution is defined in the interval $[0, 1]$; 2) Asymmetry, the peak and FWHM can be controlled by parameters $\alpha$ and $\beta$. These two characteristics make beta distribution suitable to describe the location, shape and visible pattern of occluded pedestrians. Parameterized Beta Representation is generated from the two annotated boxes, *i.e.*, full-body and visible boxes. Considering bounding box is a 2D representation and it is always axis-aligned, we utilize two independent beta distributions on the x-axis and y-axis respectively.

As mentioned before, we take the full-body box as the boundary of 2D beta distribution, while the peak along with FWHM depends on the relation between the visible part and full-body box. However, the transition relation between the peak, FWHM and the parameters $\alpha, \beta$ is hard to formulate. Instead, we calculate the mean and variance of the beta distribution with different weights assigned to the visible part and non-visible part, formulated as follows:

$$\mu_x = \frac{\int_{l_f}^{r_f} x f(x) dx}{\int_{l_f}^{r_f} f(x) dx}, \qquad \sigma_x^2 = \frac{\int_{l_f}^{r_f} (x - \mu_x)^2 f(x) dx}{\int_{l_f}^{r_f} f(x) dx},$$

$$\mu_y = \frac{\int_{t_f}^{b_f} y f(y) dy}{\int_{t_f}^{b_f} f(y) dy}, \qquad \sigma_y^2 = \frac{\int_{t_f}^{b_f} (y - \mu_y)^2 f(y) dy}{\int_{t_f}^{b_f} f(y) dy}, \qquad (3)$$

where $[l_f, t_f, r_f, b_f], [l_v, t_v, r_v, b_v]$ denote the full-body box and visible box respectively, and $f(x)$ is defined as the weight of each pixel based on the visibility:

$$f(x) = \begin{cases} W_v, \ l_v \le x \le r_v \\ W_f, \ others \end{cases}, \qquad f(y) = \begin{cases} W_v, \ t_v \le y \le b_v \\ W_f, \ others \end{cases}, \qquad (4)$$

where $W_f = 0.04, W_v = 1$ in our experiments and the size of visible box can be approximated as $w_v = \rho\sigma_x, h_v = \rho\sigma_y$ ($\rho = \sqrt{12}$). Finally, we can calculate the parameters $\alpha, \beta$ according to the normalized mean and variance, while $\lambda$ (set to $\rho/4$) is a constant to keep $\alpha, \beta > 1$:

$$\overline{\mu}_x = \frac{\mu_x - l}{r - l}, \qquad\qquad \overline{\mu}_y = \frac{\mu_y - t}{b - t},$$

$$\overline{\sigma}_x = \frac{\lambda \cdot \sigma_x}{r - l}, \qquad\qquad \overline{\sigma}_y = \frac{\lambda \cdot \sigma_y}{b - t},$$

$$\nu_x = \alpha_x + \beta_x = \frac{\overline{\mu}_x(1 + \overline{\mu}_x)}{\overline{\sigma}_x^2} - 1, \qquad \nu_y = \alpha_y + \beta_y = \frac{\overline{\mu}_y(1 + \overline{\mu}_y)}{\overline{\sigma}_y^2} - 1, \qquad (5)$$

$$\alpha_x = \overline{\mu}_x\nu_x = \overline{\mu}_x\left(\frac{\overline{\mu}_x(1 + \overline{\mu}_x)}{\overline{\sigma}_x^2} - 1\right), \qquad \alpha_y = \overline{\mu}_y\nu_y = \overline{\mu}_y\left(\frac{\overline{\mu}_y(1 + \overline{\mu}_y)}{\overline{\sigma}_y^2} - 1\right),$$

$$\beta_x = (1 - \overline{\mu}_x)\nu_x, \qquad\qquad \beta_y = (1 - \overline{\mu}_y)\nu_y.$$

Generally speaking, for each pedestrian, Beta Representation is parameterized by eight parameters, *i.e.*, $[l, t, r, b, \alpha_x, \beta_x, \alpha_y, \beta_y]$, where $[l, t, r, b]$ are the boundaries indicating the location on the image, and $[\alpha_x, \beta_x, \alpha_y, \beta_y]$ are the shape parameters of the 2D beta distribution describing the visibility of pedestrians. The probability density function of the 2D beta distribution over the whole image is formulated as follows:

$$P(x, y) = \begin{cases} C \cdot Be(\overline{x}; \ \alpha_x, \beta_x) \cdot Be(\overline{y}; \ \alpha_y, \beta_y), \ l \le x \le r, t \le y \le b, \\ 0, \qquad others, \end{cases} \qquad (6)$$

where $\overline{x} = (x - l)/(r - l), \overline{y} = (y - t)/(b - t)$, and $C$ is a normalization factor to keep the sum of PDF to 1. For pixels inside the beta boundary, the probability values are consistent with the product of two one-dimensional beta distribution, otherwise the probability values are set to zeros.

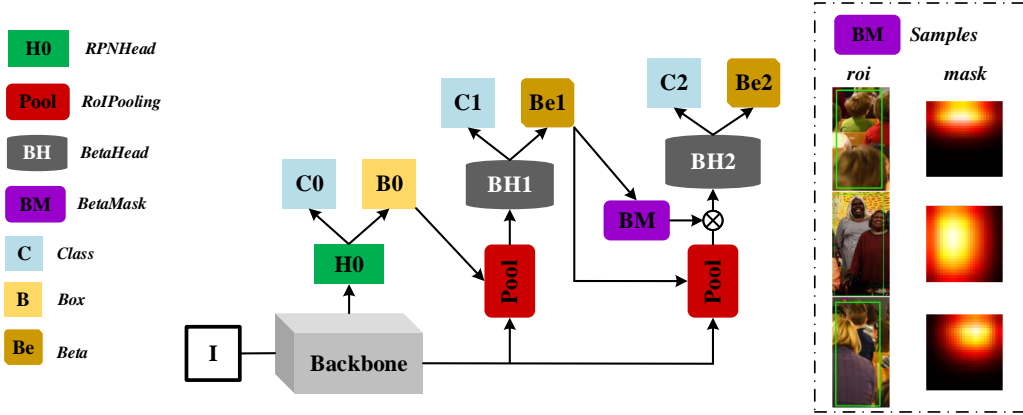

Figure 3: Beta R-CNN equipped with BetaHead and BetaMask. BetaHead regresses the class label and eight new parameters of Beta Representation, while BetaMask modulates the pooled features with beta-distribution-based masks.

### 3.1.3 Advantages

Our proposed Beta Representation shows several impressive advantages. Firstly, it is more precise in terms of the shape and visibility of pedestrians compared with box representation. While the bounding box models the object as a uniform distribution inside the box, 2D beta distribution concentrates more on the center of visual mass. Secondly, compared with the paired boxes, *i.e.*, full-body box along with visible box, 2D beta distribution treats the pedestrian more like an integrated whole rather than two individual parts. Last, it can handle a few problematic situations such as identifying highly-occluded and highly-overlapped objects, which will be discussed in detail. Moreover, it is worth mentioning that pixel-wise annotations in segmentation can also be transformed to the parameterized Beta Representation based on the above equations.

## 3.2 Beta R-CNN

To better implement the Beta Representation, we introduce a new detector named Beta R-CNN inspired by Faster R-CNN [4] and Cascade R-CNN [28]. The architecture is shown in Fig. 3. BetaHead and BetaMask are two core modules in Beta R-CNN. In the following section, we will discuss them respectively.

### 3.2.1 BetaHead

Since we adopt Beta Representation to describe a pedestrian, BetaHead is designed to regress the eight beta parameters, *i.e.*, $[l, t, r, b, \alpha_x, \beta_x, \alpha_y, \beta_y]$, which is analogous to the regression head in vanilla Faster R-CNN. Specifically, as $\alpha, \beta$ are too abstractive to learn, we adopt the mean and variance as regression targets, *i.e.*, $[l, t, r, b, \mu_x, \mu_y, \sigma_x, \sigma_y]$. The four boundary parameters, *i.e.*, $[l, t, r, b]$, utilize the same normalization strategy introduced in [4]. And for the other four shape parameters, *i.e.*, $[\mu_x, \mu_y, \sigma_x, \sigma_y]$, we adopt the normalization as follows:

$$
\begin{aligned}
t_{\mu_x} &= (\mu_x - x_a)/w_a, & t_{\mu_y} &= (\mu_y - y_a)/h_a, \\
t_{\sigma_x} &= log(\sigma_x/w_a), & t_{\sigma_y} &= log(\sigma_y/h_a), \\
t_{\mu_x}^* &= (\mu_x^* - x_a)/w_a, & t_{\mu_y}^* &= (\mu_y^* - y_a)/h_a, \\
t_{\sigma_x}^* &= log(\sigma_x^*/w_a), & t_{\sigma_y}^* &= log(\sigma_y^*/h_a),
\end{aligned}
\tag{7}
$$

where $x, y, w, h$ denote the center coordinates and size of the boundary; $\mu_x, \sigma_x, \mu_y, \sigma_y$ denote the mean and variance of the object; $\mu$ and $\mu^*$ stand for the predicted and ground-truth beta respectively, while subscript $a$ denotes the anchor box. SmoothL1 loss is adopted to optimize the BetaHead.

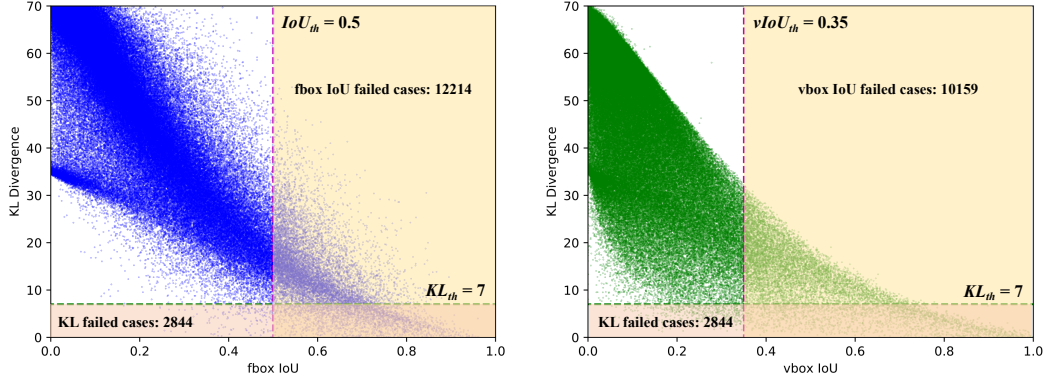

Figure 4: Comparisons between symmetrized KL divergence and IoU on CrowdHuman validation set. Each dot is a pair of two overlapped pedestrians in the same scene, measured by symmetrized KL divergence between their Beta Representation and IoU-based on full-body/visible box. The number of total dots is 206088. (The thresholds are tested in experiments.)

### 3.2.2 BetaMask

BetaMask is another novel module introduced in Beta R-CNN. Most pedestrian detectors treat the whole extracted features of a person equally important, which will result in poor performance for high-occluded scenes due to the obvious noise. As we introduced in Sec. 3.1, Beta Representation itself has different focuses to picture a person, which emphasizes the visible part in occluded scenes. It is very intuitive to adopt attention mechanism with 2D beta distribution to highlight the features of visible parts and suppress other noise simultaneously, which could induce the network to pay more attention to the discriminative features and achieve better localization accuracy and higher confidence.

Different from the common attention mechanism, our proposed BetaMask is based on 2D beta distribution, which is more targeted. In this paper, we directly generate the mask based on prediction results of the previous BetaHead instead of a CNN module like [16], as the beta mask is more like a parameterized probability distribution and it is difficult to keep the consistency of the distribution with convolutional kernels. Referring to equation (5), we get $[\alpha_x, \beta_x, \alpha_y, \beta_y]$ from the predicted $[l, t, r, b, \mu_x, \mu_y, \sigma_x, \sigma_y]$, and the mask values are sampled from the 2D beta distribution $Be(x, y; \alpha_x, \beta_x, \alpha_y, \beta_y) = C \cdot Be(x; \alpha_x, \beta_x) \cdot Be(y; \alpha_y, \beta_y)$. Then we utilize the element-wise product to modulate the pooled feature with sampled beta masks. Finally, we use KL divergence as the loss function to supervise the BetaMask module:

$$L_{mask} = \Sigma Be^*(x, y)(log Be^*(x, y) - log Be(x, y)), \tag{8}$$

where $Be^*(x, y)$ refers to the distribution generated from the ground truth, while $Be(x, y)$ is generated from the predicted beta parameters.

### 3.3 BetaNMS

When it comes to NMS, instead of taking IoU as the metric to measure the difference between detected objects, we follow [27] to utilize KL divergence as an alternative, but based on 2D beta distribution rather than bivariate normal distribution in [27]. KL divergence is defined as follows:

$$D_{KL}(p||q) = \sum_{x,y} p(x, y)(log(p(x, y)) - log(q(x, y))), \tag{9}$$

where p and q refer to two parameterized distributions. In practice, to keep the symmetry of the distance metric, we adopt the symmetrified KL divergence $\bar{D}_{KL}(p||q)$ as:

$$\bar{D}_{KL}(p||q) = (D_{KL}(p||q) + D_{KL}(q||p))/2. \tag{10}$$

Figure. 4 shows significant differences between symmetrized KL divergence metric and IoU metric on the CrowdHuman validation set. Each dot stands for a pair of two overlapped (fIoU > 0) pedestrians in the same scene, while there are 206088 dots in each graph. When we adopt KL divergence and

IoU to perform non-maximum suppression between the above paired boxes respectively, we find only 2844 failed cases based on KL divergence while there are more than 10000 failed cases based on IoU neither fIoU nor vIoU. The comparisons actively demonstrate the superiority of our proposed Beta Representation and the BetaNMS strategy. More details will be shown in experiments.

# 4 Experiment

## 4.1 Datasets

**CityPersons Dataset.** The CityPersons dataset [2] is a subset of Cityscapes which only consists of person annotations. There are 2975 images for training, 500 and 1575 images for validation and testing. The average number of pedestrians in an image is 7. We evaluate our proposed method under the full-body setting, following the evaluation protocol in [2], and the partition of validation set follows the standard setting in [19] on account of visibility: Heavy $[0, 0.65]$, Partial $[0.65, 0.9]$, Bare $[0.9, 1]$, Reasonable $[0.65, 1]$.

**CrowdHuman Dataset.** The CrowdHuman dataset [1], has been recently released to specifically target the crowd issue in the human detection task. There are 15000, 4370, and 5000 images in the training, validation, and testing set respectively. The average number of persons in an image is 22.6, which is much more crowded than other pedestrian datasets. All the experiments are trained on the CrowdHuman training set and evaluated on the validation set.

**Evaluation Metric.**
$AP$ (Averaged Precision), which is the most popular metric for detection. AP reflects both the precision and recall ratios of the detection results. Larger AP indicates better performance.
$MR^{-2}$, which is short for log-average Miss Rate on False Positive Per Image (FPPI) in [29], is commonly used in pedestrian detection. Smaller $\mathrm{MR}^{-2}$ indicates better performance. $\mathrm{MR}^{-2}$ emphasizes FP and FN more than AP, which are critical in pedestrian detection.

## 4.2 Implementation Details

In this paper, we adopt Feature Pyramid Network (FPN) [30] with ResNet-50 [31] as the backbone for all the experiments. The two-stage Cascade R-CNN [28] is taken as our baseline detection framework to perform coarse-to-fine mechanism for more accurate beta prediction. As for anchor settings, we follow the same anchor scales in [30], while the aspect ratios are set to $\mathrm{H:W} = \{1:1, 2:1, 3:1\}$. For training, the batch size is 16, split to 8 GPUs. Each training round includes 16000 iterations on CityPersons and 40000 iterations on CrowdHuman. The learning rate is initialized to 0.02 and divided by 10 at half and three-quarter of total iterations respectively. During training, the sampling ratio of positive to negative proposals for RoI branch is $1:1$ for CrowdHuman and $1:4$ for CityPersons. On CityPersons, the input size for both training and testing is $1024 \times 2048$. On CrowdHuman, the short edge of each image is resized to 800 pixels for both training and testing. It is worth mentioning that the proposed components like BetaHead in Beta R-CNN are all optimization-friendly, thus there is no essential difference between Beta R-CNN and Faster R-CNN [4] or Cascade R-CNN [28] for model training and testing.

## 4.3 Ablation Study on CrowdHuman

**Ablation study and main results.** Table 1 shows the ablation experiments of the proposed Beta R-CNN in Sec. 3, including BetaHead, BetaMask, Mask Loss, and BetaNMS. The baseline is a two-stage Cascade R-CNN with default settings introduced in Sec. 4.2. As we claimed in Sec. 3, it is clear that our method consistently improves the performance in all criteria. BetaHead and BetaMask are proposed to implement Beta Representation and alleviate the occluded issue with new regression targets and attention mechanism, which surely reduce the $\mathrm{MR}^{-2}$ from $43.8\%$ to $41.3\%$ and improve AP from $85.2\%$ to $87.1\%$. And the Mask Loss, *i.e.*, equation 8, helps model get a more accurate mask. Moreover, the improvement of BetaNMS well demonstrates the superiority over the IoU-based NMS. We further analyze the role of each module. Beta Representation could picture more details of the shape and visibility of pedestrians especially in occluded and crowded scenes, and BetaMask adopts attention mechanism by utilizing 2D beta distribution to modulate more discriminative features, which enhances Beta R-CNN further. At last, BetaNMS eliminates the

inherent drawback of IoU-based NMS when it meets highly-overlapped instances under crowded scenes. More details can be found in Sec. 3.

Table 1: Ablation Study on CrowdHuman

| BetaHead | BetaMask | MaskLoss | BetaNMS | $\text{MR}^{-2}$/% | AP/% |
|---|---|---|---|---|---|
|  |  |  |  | 43.8 | 85.2 |
| ✓ |  |  |  | 43.5 | 85.5 |
| ✓ | ✓ |  |  | 41.3 | 87.1 |
| ✓ | ✓ | ✓ |  | 41.1 | 87.2 |
| ✓ | ✓ | ✓ | ✓ | **40.3** | **88.2** |

**Comparison with various NMS strategies.** To powerfully illustrate the effectiveness of the BetaNMS, we compare BetaNMS with IoU-based NMS on full-body/visible boxes (visible boxes are approximately transformed from Beta Representation). Results are shown in Table 2 and all reported experiments here are based on Beta R-CNN. BetaNMS outperforms all other NMS methods with a large margin. Compared with fIoU-, vIoU-based NMS tends to recall more overlapped instances but bring in more false positives meanwhile, reflecting in the higher $\text{MR}^{-2}$ and AP. In addition, although we integrate fIoU and vIoU in NMS, we can find BetaNMS still outperforms by at least 0.4% on $\text{MR}^{-2}$ and 1.5% on AP, which means BetaNMS surely better distinguishes highly overlapped instances than IoU-based NMS, whether it is based on the full-body box or visible box or both.

Table 2: Comparisons of various NMS strategies

| Strategy | Threshold | $\text{MR}^{-2}$/% | AP/% |
|---|---|---|---|
| fIoU NMS | 0.5 | 41.1 | 87.2 |
| vIoU NMS | 0.35 | 42.0 | 88.1 |
| fIoU + vIoU NMS | 0.5/0.35 | 41.0 | 88.1 |
| SoftNMS | - | 41.1 | 88.0 |
| BetaNMS | 6 | 40.6 | **89.6** |
| BetaNMS | 7 | **40.3** | 88.2 |

**Speed/accuracy trade off.** Each proposed module in Beta R-CNN is light-weight with little computation cost. We take CrowdHuman validation set with 800x1400 input size to conduct speed experiments on NVIDIA 2080Ti GPU with 8 GPUs, and the average speeds are 0.483s/image ( *Cascade R-CNN baseline*) and 0.487s/image (*Beta R-CNN*) respectively. The difference can be negligible.

## 4.4 State-of-the-art (SOTA) Comparison on CrowdHuman

Comparisons with some recent methods on the CrowdHuman validation set are shown in Table 3. It clearly shows that our Beta R-CNN outperforms others with a large margin, especially on the metric $\text{MR}^{-2}$. Such a large gap demonstrates the superiority of our Beta R-CNN. It is worth noting that CrowdDet [32] achieves a little higher AP than ours, which attributes to its motivation, *i.e.*, laying emphasis on larger recall at the expense of more false positives, reflecting in higher $\text{MR}^{-2}$ than ours.

Table 3: SOTA comparisons on CrowdHuman

| Methods | $\text{MR}^{-2}$/% | AP/% |
|---|---|---|
| CrowdHuman [1] | 50.4 | 85.0 |
| Adaptive NMS [22] | 49.7 | 84.7 |
| PBM [18] | 43.3 | 89.3 |
| CrowdDet [32] | 41.4 | **90.7** |
| Beta R-CNN($KL_{th} = 7$) | **40.3** | 88.2 |
| Beta R-CNN($KL_{th} = 6$ ) | 40.6 | 89.6 |

Table 4: SOTA comparisons on CityPersons

| Methods | Reasonable | Heavy | Partial | Bare |
|---|---|---|---|---|
| OR-CNN [8] | 12.8 | 55.7 | 15.3 | 6.7 |
| TLL [33] | 15.5 | 53.6 | 17.2 | 10.0 |
| RepLoss [19] | 13.2 | 56.9 | 16.8 | 7.6 |
| ALFNet [34] | 12.0 | 51.9 | 11.4 | 8.4 |
| CSP [35] | 11.0 | 49.3 | 10.4 | 7.3 |
| Beta R-CNN | **10.6** | **47.1** | **10.3** | **6.4** |

### 4.5 Experiments on CityPersons

To further verify the generalization ability of our method, we also conduct experiments on CityPersons. Table 4 compares Beta R-CNN with some state-of-the-art methods. For a fair comparison, we only list those methods that follow the standard settings, *i.e.*, adopting subset partition criterion in [19] and feeding images with original size as inputs when performing evaluation. Because of the space limit, we will report the results with 1.3x enlarged input images in our supplementary materials. From the table, we can see that our Beta R-CNN outperforms all published methods on all four subsets, especially with a large margin on the Heavy subset, which verifies that our method is effective in occluded and crowded scenes.

## 5 Conclusion

In this paper, we propose a statistic representation for occluded pedestrians based on 2D beta distributions, which takes the paired boxes as an integrated whole and emphasize the object center of visual mass. Besides, Beta R-CNN, equipped with BetaHead and BetaMask, aims to alleviate the pedestrian detection in occluded and crowded scenes. BetaNMS could effectively distinguish highly-overlapped instances based on Beta Representation and KL divergence. The quantitative and qualitative experiments powerfully demonstrate the superiority of our methods. Beta Representation, as well as BetaHead, BetaMask, BetaNMS are all flexible enough to be integrated into other two-stage or single-shot detectors and are also compatible with existing optimization methods to further boost their performance. Moreover, our method could be extended to more general scenes and other detection tasks.

## Acknowledgements

This work was supported in part by the National Key Research and Development Program of China under Grant 2016QY02D0304 and the National Natural Science Foundation of China under Grant 60572002.

## Broader Impact

Our contributions focus on the novel representation and pipeline for pedestrian detection, which can be extended to other computer vision tasks. Also, it may provide new ideas for follow-up research. It therefore has the potential to advance both the beneficial and harmful applications of object detectors, such as autonomous vehicles, intelligent video surveillance, robotics and so on. As for ethical aspects and future societal consequences, this technology can bring harmful or beneficial effects to the society, which depends on the citizens who have evil or pure motivation and who can make good use of this technological progress.

## Footnotes

[2]FIoU and vIoU are the IoU calculated based on full-body/visible boxes respectively.

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
