[Supplementary Material]

# Supplementary Material

## 1  Comparisons on CityPersons

As some SOTA methods also report results with $1.3\times$ enlarged input size on CityPersons, here we evaluate our Beta R-CNN with both original and enlarged input size. The results are shown in Table 1. Due to the space limit in the paper, some baseline methods are also reported in the table for a thorough comparison. By the way, we have to emphasize again that we only list the methods which follow the standard settings in RepLoss[19] for a fair comparison. From the table, we can see that whether with $1.3\times$ enlarged or original input size, our method both outperforms other state-of-the-art methods with a large gap, especially on Heavy subset.

Table 1: Comparisons on CityPersons

| Methods | Size | Reasonable | Heavy | Partial | Bare |
|---|---|---|---|---|---|
| Adapted Faster RCNN [2] | $\times1$ | 15.4 | - | - | - |
| OR-CNN [8] | $\times1$ | 12.8 | 55.7 | 15.3 | 6.7 |
| RepLoss [19] | $\times1$ | 13.2 | 56.9 | 16.8 | 7.6 |
| Adaptive NMS [22] | $\times1$ | 12.0 | 51.2 | 11.9 | 6.8 |
| TLL [33] | $\times1$ | 15.5 | 53.6 | 17.2 | 10.0 |
| TLL + MRF [33] | $\times1$ | 14.4 | 52.0 | 15.9 | 9.2 |
| ALFNet [34] | $\times1$ | 12.0 | 51.9 | 11.4 | 7.3 |
| CSP [35] | $\times1$ | 11.0 | 49.3 | 10.4 | 7.3 |
| **Beta R-CNN** | $\times1$ | **10.6** | **47.1** | **10.3** | **6.4** |
| Adapted Faster RCNN [2] | $\times1.3$ | 12.8 | - | - | - |
| OR-CNN [8] | $\times1.3$ | 11.0 | 51.3 | 13.7 | **5.9** |
| RepLoss [19] | $\times1.3$ | 11.6 | 55.3 | 14.8 | 7.0 |
| Adaptive NMS [22] | $\times1.3$ | 10.8 | 54.0 | 11.4 | 6.2 |
| CrowdDet [32] | $\times1.3$ | 10.7 | - | - | - |
| **Beta R-CNN** | $\times1.3$ | **9.9** | **45.8** | **9.1** | 6.0 |

## 2  Visualization

To show the performance of our proposed Beta R-CNN more intuitively, we visualize the results of several images with high-occluded scenes from the CrowdHuman, which are shown in Fig. 1. We can find that although people in these images overlap with each other very seriously, Beta R-CNN still detects and distinguishes them very well, benefiting from Beta Representation and BetaNMS. Besides, we randomly draw the corresponding Beta Representation over the last image for better understanding.

Figure 1: Visualization results of Beta R-CNN on CrowdHuman.