[Reviews · NeurIPS 2020]

Review 1

Summary and Contributions: The paper proposes a representation called “Beta Representation” as opposed to bounding box based representation for detecting pedestrians from a crowded scene. The new representation is based on 2D beta distributions with 8 parameters (two sets of alpha and beta parameters for the 2D beta distributions as well as four parameters to localize the pedestrian). The representation models a pedestrian with a pixel-level map with the traditional visible and full body bounding boxes implicitly represented in the parameters of the 2D beta distribution. The main advantage of beta representation supposedly is its loosening of the constraint purely full-body bounding boxes impose on the object representation, i.e. the assumption that the object is mostly visible. Hence, this new pixel-level representation could potentially alleviate the drawbacks of full body bounding box representation in the presence of extreme occlusion that is typical of crowded scenes. A specific implementation of this representation was performed using a new R-CNN model by introducing two components, i.e. beta head (for regression of the 8 2D beta parameters) and beta mask (to introduce 2D beta masks). Furthermore, the authors also replaced the common non-maximal suppression (NMS) with a beta version to avoid highly occluded pedestrians from being suppressed in the regular NMS in post-processing. For comparison of two beta representations, the authors use KL divergence as metric as opposed to IoU, which is common in comparing two bounding boxes. The authors performed experiments on two standard datasets (CityPersons and CrowdHuman) that contain crowded pedestrian scenes to show the advantage of their representation over traditional bounding box detection.

Strengths: - 2D axis aligned bounding box based representation have several drawbacks and had implicit assumptions that might break in real-world scenarios. Bounding box representation assumes the relative orientation of the camera with respect to the pedestrian under consideration is unchanged and mostly assumed the pedestrian could be axis-aligned. However, in real-world situations such as network of camera placements, several pedestrian positions such as sitting and leaning in different poses, and occlusions would make it difficult to represent the pedestrian with an axis-aligned bounding boxes. The proposed representation has the potential to tightly fit to the pedestrian and could potentially help in severe occlusions. - They clearly articulated the problem, the existing bounding box solutions, their drawbacks, and the advantages of the 2D beta representation. - The selection of performance metrics is sound to reflect balanced performance and fair comparison with existing methods. - The ablation study seems sound testing the progressive inclusions of each of the new/modified components.

Weaknesses: - The title is too broad although the paper is proposing this new representation. It is essentially defining what “Beta Representation” is. However, the paper is proposing a specific implementation using specific R-CNN model. Hence, I advise changing the tittle to reflect the scope of the specific implementation and experiments performed. This is especially useful as R-CNN is not state-of-the art detector anymore for a variety of reasons. Hence, to distinguish this work from future works that could employ more advanced detection model that will build on top of this work, the title should be a bit scoped despite the introduction of the representation in this work. Generally 2D beta representation is not completely a new concept. It is used in hand trajectory representation among other tasks. - In the larger dataset (CrowdHuman) the lack of separate test set and their evaluation on the validation set is limiting. To the authors defense this could be due to the dataset inherent split and hence for making comparison with existing methods it might makes sense to use the existing split. However, the authors could also attempt to perform their own split with complete training, validation, and test set and reevaluate the existing methods using this split for fair comparisons.

Correctness: - The experimental design, the selected datasets (based on their approperiateness to highlight the advantages of the proposed method), and the selection of performance metrics are all sound. However, omission of some existing methods because they are trained under a slightly different condition such as image scaling makes it hard to draw strong conclusions on the benefits of the proposed model. - Given the tight fit of the 2D beta representation compared to 2D axis-aligned bounding boxes and the purported advantages of beta representation in such highly crowded scenes datasets with multiple severe occlusions, one would expect to see significant performance improvement over existing methods. However, the improvements in tables 3 and 4 are not that convincing.

Clarity: - The paper is written well and is easy to read. It could use some proof reading for clarity. Also, authors should have it proof-read for grammatical correctness in some places. - On line 212, please define AP for clarity. “AP, which is the most popular metric for detection.” - On Table 3 and 4, please either define SOTA or define it where the tables are first referenced in line 248. - Tables 4 caption doesn’t make it obvious the metric displayed is MR^-2. Generally the table and figure captions could be improved be very specific and more descriptive throughout the paper.

Relation to Prior Work: - There is a dedicated related work section. The problem is correctly formulated in the context of existing pedestrian detection methods. - The selection of the balanced metrics for comparison with existing methods is appropriate. However, omission of some existing methods makes it harder to draw strong conclusion. - The selection of R-CNN as the base model to implement the proposed representation also makes it harder to compare it with existing detectors based on advanced object detectors.

Reproducibility: Yes

Additional Feedback:


Review 2

Summary and Contributions: This paper proposed Beta Representation for pedestrian detection. Combined with BetaNMS, Beta Representation is much better for distinguishing highly-overlapped instances in crowded scenes. This module is validated on the CityPersons and CrowdHuman datasets.

Strengths: The proposed Bate Representation is a novel representation for pedestrian, which has reached state-of-the-art results on CrowdHuman and CityPersons datasets. KL divergence is used instead of IoU to measure the distance between two objects and proposed new NMS strategy BetaNMS, and verified its effectiveness in ablation experiments.

Weaknesses: 1. In the ablation study on CrowdHuman, the MaskLoss did not play an effective role, whether to consider designing other loss functions, or remove. 2. In the figure 3, the output of the Be module is confusing. The visualization shows the mask of the BM module, but the output of the Be module is not explained clearly. 3. How to calculate the relation between the visible part and the full-body requires a more detailed explanation, and a full analysis of the reasons for the effectiveness of this relation. 4. For the BetaNMS module, it is recommanded that to add a comparative test using softNMS.

Correctness: Nothing to correct.

Clarity: The structure of the whole article is reasonable. The method part describes the implementation, but the inner meaning and motivation are not explained clearly.

Relation to Prior Work: Compared with previous work, but not enough.

Reproducibility: Yes

Additional Feedback: In the auther’s feedback, compared the effectiveness of softNMS and BetaNMS, and BetaNMS performed better. But the reason for the performance of BetaNMS, a simple mathematical explanation needs to be given. Beta Representation utilizes 8 parameters to identify a pedestrian, and the 8 parameters are generated from annotated full-body boxes and visible boxes, without visible boxes label, this method cannot be used, so this method is not scalable. In addition to AP and MR, other performance indicators, such as training time, should also be explained briefly.


Review 3

Summary and Contributions: To deal with the pedestrian detection in crowded scenes, the authors propose to use 2D beta distribution to model the full-body box and visible box. The proposed Beta Representation is further used in the new NMS method, which removes the duplicate predictions according to the KL divergence. The authors validate their method on the mainstream and challenging CrowdHuman and CityPersons datasets.

Strengths: - The main idea of using the beta distribution to construct the relationship of full-body and visible box is novel and interesting. - I especially like the solution of BetaNMS, which shows successful detections while the original IoU-based NMS fails. As we know, the NMS is a main obstacle for detecting pedestrians in a crowd. - The results on CrowdHuman and CityPersons datasets are good.

Weaknesses: 1. There is no speed or cost report which can show the overhead of each component and the final speed/accuracy trade-off. 2. The authors use a cascade framework, as in Fig3. They should clarify that wether the improvements come from the cascade design. 3. Lack of analysis of how each component contributes, e.g., the BetaHead, BetaMask and BetaLoss in Table 1. 4. The above concerns make me confused about the soundness of the claims.

Correctness: The claims can be further validated.

Clarity: The paper is clear but not good enough. I list some comments below.

Relation to Prior Work: Yes, the related work is clearly reviewed.

Reproducibility: Yes

Additional Feedback: - The figures should be refined, especially the coarse Fig 4. - The notations should be clearly illustrated, e.g., those in section 3.3. - The bold '89.6' in Table 3 is not correct.


Review 4

Summary and Contributions: This paper concerns pedestrian detection in occluded and crowded scenes. They observe that conventional representation of bounding box (x1,x2; y1,y2) is limited and poor approximation of location and size. To this end, beta representation is proposed against the conventional bounding box representation. With this they propose Beta RCNN and BetaNMS. Results are positive.

Strengths: There are several things to like about this paper: 1. Pedestrian detection in crowded/occluded scenes are very challenging. Seeing progress towards this direction is essential. 2. The way the authors formulate new representation is interesting. 3. The state-of-the-art performance.

Weaknesses: I list below the unclear parts and weaknesses: (1) Requirement: Does the proposed approach require two kinds of annotations to generate beta representation, one full body, another for visible body? (2) Network structure: This could be explained better. Why the authors design in this way? Why are there two BH heads, but only one BM head? (3) Evaluation: Is there a connection between MR^{-2} and AP. In my understanding, there is a strong correlation, that is, if MR-2 decreases AP should improve. Which metric is more suitable for pedestrian detection? (4) Practical: I understand that the results of this paper is the state-of-the-art. However, how practical is the proposed approach: a) Is training easier than Faster RCNN? b) In Table 2, to my eyes performances are very similar, why should one choose BetaNMS? c) If BetaNMS is integrated into EMD [32], will the performance of that improve further? c) Can the proposed method be integrated to a single-shot detector? d) Is beta-representation applicable to just standard pedestrian detection eg. Caltech pedestrian dataset? It would be great if the authors consider this.

Correctness: As far as my knowledge concerned the technical content of the paper appears to be correct.

Clarity: Overall yes. Related work should be improved by discussing state-of-the-art methods.

Relation to Prior Work: No. As authors mentioned [27] is the most related work. However, it was not discussed/compared in detail. In fact, I glanced [27] and it is easy to see that there are several things that are similar eg., KL. Since authors claim state-of-the-art methods, it would be great to discuss state of the art methods in related work section. Those methods are mentioned briefly in Experiments section.

Reproducibility: Yes

Additional Feedback: One should have a strong understanding of Faster RCNN and detector concepts. Please see Weaknesses section. Thanks. -- After Rebuttal -- I would like to keep my original score after reading rebuttal and the other reviewers' comments.

[Author Response · NeurIPS 2020]

We thank all reviewers for their thoughtful comments. Below we integrate all the concerns and clarify them in details.

**(R1) Omission comparisons with SOTA methods.** As we have claimed in Line 258, we have reported almost all
published SOTA methods under a fair comparison in our supplementary material due to the space limit, regardless of
feeding images with original or 1.3x enlarged size. Our Beta R-CNN outperforms all of them especially on crowd
scenes. We boost SOTA results by $\mathbf{1.1\%}\ \boldsymbol{MR^{-2}}$ on CrowdHuman, $\mathbf{2.2\%}$ and $\mathbf{5.5\%}\ \boldsymbol{MR^{-2}}$ on Heavy subset of
CityPersons with original and 1.3x enlarged size respectively. Considering the datasets' challenge and other SOTA
methods' improvements like [32, 22], our improvement is really convincing. Besides, we thank R1 for raising our
concerns about performing our own split on CrowdHuman, it is good advice, and we will take it in the final version.

**(R1, R2, R3, R4) Questions about ablation study. Firstly**, we thank R3 for the reminding about insufficient analyses
of each component in Sec.4.3. Actually we have claimed and validated the effect of each module in Sec.3 like Sec.3.3
for BetaNMS, so we mainly focus on the results to verify their effectiveness in Sec.4.3. We will polish our paper
writing more reasonably. **Secondly**, we adopt R2's advice to add a comparative test using softNMS, and the result is
$41.1\%\mathrm{MR}^{-2}, 88.0\%\mathrm{AP}$, which is inferior to our method. **Thirdly**, although the improvement of MaskLoss is not as
impressive as other modules, but it is proposed to better implement Beta Representation, and actually we have tried
other loss functions to supervise Beta Mask but failed to further improve the performance except MaskLoss. We will
explore to boost MaskLoss in future work. **Besides**, actually R-CNN models are widely used by most SOTA pedestrian
detectors [8,19,22,32]. Comparisons with advanced detectors will be explored in the future work. **Last**, Table 1 clearly
shows that although our cascade baseline achieves very high results, our method still achieves $\mathbf{3.5\%}\ \boldsymbol{MR^{-2}}$ and **3.0%**
$\boldsymbol{AP}$ gains, which can verify the effectiveness of our method and well clarify R3's concerns about our improvements.

**(R2, R4) Questions about model design.** Beta R-CNN is based on Cascade R-CNN by replacing the box regression
head with our BetaHead (shown in Fig.3 as Be) to regress eight Beta parameters, which has been introduced in Sec.3.2.1.
Besides, BetaMask is proposed to further improve the performance. But due to the coarse results from RPN, BetaMask
is hard to contribute to the first BetaHead, thus we add one BetaMask between two adjacent BetaHeads at last.

**(R2, R4) Questions about Beta Representation.** Beta Representation utilizes 8 parameters to identify a pedestrian,
and the 8 parameters are generated from annotated **full-body boxes and visible boxes**. As introduced in Sec 3.1.2, the
8 parameters include 4 boundary parameters and 4 shape parameters. Boundary parameters are consistent with the
full-body box, while shape parameters control the peak and width of 2D beta distribution. For human visual habits, we
tend to emphasize the visible parts, so we assign different weights for visible/non-visible parts and calculate the mean $\mu$
and variance $\sigma$. Based on $\mu$ and $\sigma$, we could deduce the shape parameters according to the beta distribution properties.
Then, the 2D beta distribution could highlight the visible part and suppress the non-visible part.

**(R2, R4) Comparisons with related works.** Due to space limit, our descriptions of related works in Sec.2 maybe a
bit concise, we will add more details in the final version. Besides, R4 mentioned $Object\ as\ Distribution$ [27]. The
similarity between [27] and our method is that we both adopt distribution to represent objects, but actually there are
many differences. [27] utilizes the bivariate normal distribution and we have claimed its weaknesses in Line 94-97. [27]
can't model the unpredictable visible patterns about pedestrians. Also, [27] needs more complex training strategies and
its performance is considerably poor. As for KL, i.e., Kullback-Leibler divergence, it is a common metric to measure
the distance between two distributions, and we adapt it to our BetaNMS.

**(R3) Speed/accuracy trade-off.** Each proposed component in Beta R-CNN is light-weight with little computation cost.
We take CrowdHuman validation set with 800x1400 input size to conduct speed experiments on NVIDIA 2080Ti GPU
with 8 GPUs, and the average speeds are **0.483s/image (Cascade R-CNN baseline)** and **0.487s/image (Beta R-CNN)**
respectively. Besides, the total flops of them are 52.77G and 52.78G respectively. The difference can be negligible.

**(R4) Is Beta R-CNN practical? Firstly**, the proposed components like BetaHead in Beta R-CNN are all optimization-
friendly, so the training of Beta R-CNN is as easy as Faster R-CNN and Cascade R-CNN. **Secondly**, Beta Representation
is based on beta distribution, it is intuitive to adopt BetaNMS (based on KL) to measure distance between distributions,
which achieves further performance improvement comparing with other NMS strategies without any extra cost. **Thirdly**,
Beta Representation, as well as BetaHead, BetaMask, BetaNMS are all flexible enough to be integrated into other
two-stage or single-shot detectors, and are also compatible with existing optimization methods like EMD[32] to further
boost their performance. **Last**, Beta R-CNN is proposed for occlusion and crowd issues but it still works well on
standard scenes like Reasonable subset of CityPersons. So Beta R-CNN can also be generalized on Caltech.

**(R4) Questions about evaluation metric.** Some connection exists between $\mathrm{MR}^{-2}$ and AP, but higher AP doesn't mean
lower $\mathrm{MR}^{-2}$. $\mathrm{MR}^{-2}$ is more suitable by emphasizing FP&FN beyond AP, which is critical in pedestrian detection.

**(R1, R2, R3, R4) Paper writing.** We really appreciate all the useful suggestions about our paper writing, we will
polish our paper in the final version to fix all of them, i.e. figure out a better title, improve captions of tables and figures
(R1), refine Fig.4 (R3), clarify notations and definitions in details(R1,R3), and other common written problems.

[Meta-Review · NeurIPS 2020]

The paper proposes to use a novel 2D beta distribution to model the full-body box and visible box to deal with the pedestrian detection problem in crowded scenes, coping with occlusions. Then authors propose Beta RCNN and BetaNMS. Results are positive. Reviewers noted that although the paper is interesting the results , in terms of comparison and ablation study are fair enough to demonstrate the effectiveness of the proposed method compared to the existing methods. As well the paper is clear but not good enough. After rebuttal one reviewer made higher his/her rate and and in general all reviewers declared to be satisfied by rebuttal. Thus AC agrees with the acceptance of the paper.